# Research on a Small Oil Detection Instrument Combining Micro-Distillation Range Analyzer and Raman Spectroscopy Technology

**DOI:** 10.3390/s25072130

**Published:** 2025-03-27

**Authors:** Hao Yan, Chen Zhao, Xiuli Zuo, Jianhua Shu, Weixing Hua, Liang Guan, Kecheng Gu

**Affiliations:** Army Logistics Academy, Chongqing 401331, China; amanscut@sina.com (H.Y.); ciaczhao@mail.ustc.edu.cn (C.Z.); zuoxl_cq@126.com (X.Z.); shu13368232010@sina.com (J.S.); huaweixing@163.com (W.H.)

**Keywords:** oil, distillation, Raman spectroscopy, detection system

## Abstract

In order to achieve rapid detection of oil quality, an innovative small-scale oil joint detection system is designed in this paper. The joint detection system combines a micro-distillation instrument and Raman spectrometer module through a combination module. The basic principle is to use a Raman spectrometer to detect the distillate in the micro-distillation instrument in order to predict the properties of the oil and determine its quality. The system uses Savitzy–Golay smoothing, baseline correction, normalization processing, partial least squares, and other methods for data fitting to ensure the authenticity and comparability of the measured data. Fourteen different sources and grades of diesel oil are selected in this paper for pre-experiments to verify the data fitting effect. Two oil samples, No. 1−10^#^ diesel oil 1 with strong fluorescence interference and No. 2−10^#^ diesel oil 2 with weak fluorescence interference, were used as test objects to compare the Raman spectra of undistilled oil and distilled oil, verify the testing effect of the joint detection system, and analyze the mechanism of fluorescence interference mainly existing in the heavy components of the oil. The test results show that the joint detection system can quickly detect the quality of different grades of diesel oil, with good smoothness, obvious characteristic bands, and good data fitting. The joint detection system designed in this paper has the advantages of high sensitivity, high integration, and small size, laying the foundation for research related to distillation ranges combined with Raman spectroscopy technology.

## 1. Introduction

Diesel oil is a petroleum product, a complex mixture of hydrocarbons mainly composed of diesel fractions produced by processes such as crude oil distillation, catalytic cracking, and thermal cracking. It has a high energy density, a low fuel consumption rate, and is widely used in diesel vehicles, generators, etc. [1]. At present, there are many studies on the substitution of diesel, and researchers are trying to produce alternative diesel from waste tire pyrolysis oil, corn oil, cotton, and pumpkin seeds [2,3,4,5]. Some scholars have also attempted to improve diesel performance through ternary or quaternary blends [6,7,8]. In order to reduce the pollution of the environment caused by the use of diesel, China has separated automotive diesel from the original light diesel and aligned its specific standards with foreign clean diesel standards for vehicles. However, due to changes in the target audience, the original light diesel has also changed its name and been renamed as ordinary diesel. In practical work, the self-ignition point alone cannot fully represent the ignition performance. There are many other factors, such as fuel evaporation, viscosity, and engine compression ratio. The quality of diesel directly affects its storage and usage performance.

Distillation range is the most important quality indicator for evaluating diesel evaporation. By measuring the distillation range, the percentage of hydrocarbons and heavy fractions in crude oil can be inferred, thereby estimating the number of petroleum products such as gasoline, kerosene, and light diesel that can be produced from crude oil. By controlling the distillation conditions, the yield of the target product can be increased in the production process of petroleum products. When oil is used as engine fuel, its distillation range can be used to determine the applicable range and degree of the oil, thereby improving the effectiveness of its use. There are many studies on the rapid detection of distillation range, including near-infrared spectroscopy [9], gas chromatography high-temperature simulated distillation [10], and microwave distillation [11]. The standard for determining distillation range currently used in China is GB/T6536-2010 [12], which was revised based on ASTM D86-2007a [13]. Although this method has the advantages of simple and inexpensive instrument equipment, and easy operation, it also has disadvantages such as being time-consuming and allowing for large human errors.

Raman spectroscopy is an analytical technique based on the Raman scattering effect, which generates inelastic scattered photons through the interaction between matter and incident light and measures their energy changes to obtain molecular vibration/rotation information. It is widely used for material structure identification and non-destructive testing analysis of chemical composition in the petrochemical industry [14,15,16,17]. The Raman characteristic bands have a correlation with the vibrations of molecules or functional groups, which can be reflected in the spectrum. By identifying the characteristic bands, the molecules or functional groups contained in the measured substance can be determined. A Raman shift is usually expressed in a wavenumber, which is numerically equal to the absolute value of the difference in frequency between the scattered light and the incident light, with a range of approximately 40–4000 cm^−1^. Raman intensity can be used to represent the content of a substance, which is directly proportional to the concentration of its composition. This relationship can be used to calculate the composition and parameters of a substance. For different diesel samples, their compositions and contents vary, which can cause spectral changes. Establishing a model between spectral data and composition and using Raman spectroscopy technology allows us to predict properties such as diesel composition, substance content, and physicochemical parameters.

At present, there is relatively little research on the combination of distillation range detection and Raman spectroscopy technology. Gloerfelt-Tarp, F. et al. [18] showed the great potential in using Raman spectroscopy to provide real-time product quality data at commercial tea tree oil (TTO) distillation sites, enabling direct feedback control and process optimization. Naz, S. et al. [19] analyzed steam-extracted hemp (*Cannabis sativa* L.) essential oil (HEO) and its fractions obtained by fractional distillation using Raman spectroscopy linked with principal component analysis (PCA). This paper innovatively designs a small-scale oil joint detection system, as shown in Figure 1, which combines the micro-distillation instrument and Raman spectrometer modules through a combination module to achieve a rapid determination of diesel quality. Small-scale instrument detection is an important development direction [20,21]. The basic principle of this system is to use a Raman spectrometer to detect the distillate in the micro-distillation instrument in order to predict the properties of the oil and determine its quality. The system performs data fitting through Savitzy–Golay smoothing, baseline correction, normalization, partial least squares, and other methods [22] to ensure the authenticity and comparability of the measured data. In order to verify the feasibility of the joint detection system, 14 different sources of diesel samples were selected for experiments. Our process was to obtain the distillation range curve and Raman spectrum of the distillate during the distillation process, compare the Raman spectrum obtained during the distillation process with the Raman spectrum of the sample without distillation, and verify the testing effect of the joint detection system. The test results show that the joint detection system can quickly detect the quality of different grades of diesel, with good smoothness and obvious characteristic bands. The joint detection system designed in this paper has the advantages of high sensitivity, high integration, and small size, laying the foundation for research related to distillation range detection combined with Raman spectroscopy technology.

## 2. Design and Preparation of Instruments

### 2.1. Hardware Design

Hardware design is the foundation of the entire system and is crucial for its normal operation. The hardware design part of this paper combines the micro-distillation instrument and Raman spectrometer modules through a combination module to achieve real-time detection using the Raman spectrometer during the distillation process.

#### 2.1.1. Self-Developed Micro-Distillation Instrument

In the early stages of this paper, a miniature distillation instrument that integrates mechanical, electronic, and computer technologies was designed. It can be used for temperature control, pressure regulation, and to determine the distillation characteristics of various oil products under normal pressure. It automatically completes the entire distillation process experiment. It can be directly connected to a computer system and send measurement data to the computer program for calculation and display through communication lines. This instrument is based on the ASTM D7345 standard [23] and has the characteristics of simple operation and stable performance. Its sample dosage is 10 mL. Its heating temperature is ≤500 °C, suitable for fuel samples with boiling points of 20–400 °C. Its pressure range is the atmospheric pressure range (73.33 kPa–106.7 kPa). The test timeis ≤10 min.

#### 2.1.2. Self-Developed Raman Spectrometer

The miniaturization transformation of the Raman spectrometer is carried out independently in this paper. The front and back views of the self-developed Raman spectrometer are shown in Figure 1a,b. The instrument has been designed in proportion to reduce its size to 230.0 mm × 150.0 mm × 54.0 mm. As shown in Figure 1c, the internal design of the self-developed Raman spectrometer has been improved by the miniaturized design of the internal structure, using a 785 nm laser as the light source and a resolution of 3 cm^−1^. The maximum power is 340 mW and adjustable. The instrument is compact in size and has a simple optical path structure, making it easy to upgrade, modify, and integrate for subsequent designs. The Raman spectrometer consists of three parts: a laser, a spectral spectroscopic receiving system, and a Raman probe. The laser spot size is in the millimeter range, and signal collection is carried out through optical fibers with an integration time of 10 s. The correction method is to place the standard sample at the measurement position of the spectrometer, ensuring that the light spot is aligned with the surface of the sample. We then compare the deviation between the measured peak position and the standard peak position (the Raman characteristic peak of the silicon wafer is located at about 520 cm^−1^, and the wavenumber offset needs to be adjusted). Input theoretical values through the calibration tool in the spectrometer software, and the system automatically or manually adjusts the wavenumber offset. Calibrate once before each experiment, and do not calibrate for consecutive experiments.

#### 2.1.3. Combination Module Design

In order to achieve real-time detection of distillates by Raman spectrometer while distilling diesel samples, a combination module was designed in this paper.

The internal structural principle is shown in Figure 2. The distillate flows through the combination module, and the laser light path enters the path of the distillate. The spectroscope returns the light path, and the returned light signal is collected for Raman spectroscopy.

The inflow and outflow of distillate are comprehensively considered in this paper. Laser injection and retraction strive to miniaturize the combination module as much as possible while meeting the measurement function in order to facilitate the design and production of physical objects in the later stage. Figure 3a is a combined module outline design drawing, Figure 3b front view, Figure 3c side view, and Figure 3d top view. The physical and side views of the combined module are shown in Figure 3e,f.

### 2.2. Software Design

The software part is mainly divided into upper computer control software and lower computer software. As the Raman spectrometer uses finished instruments, the upper computer software is mainly used to control the micro-distillation instrument and Raman spectrometer to perform synchronous testing, collect data, and calculate measurement results during the testing process. The lower computer software is mainly developed for the control of the micro-distillation instrument. Software is the key factor driving hardware, and only by driving hardware through software can the designed functions be realized. The key part of the software is to simultaneously control the Raman spectrometer and micro-distillation instrument and to collect data.

#### 2.2.1. Micro-Distillation Instrument Software

The micro-distillation instrument software directly controls the instrument using an upper computer. The software controls the operation of the micro-distillation instrument, achieving four main functions: establishing a distillation range experimental plan, controlling the distillation range experimental process, collecting distillation range experimental data, and calculating and analyzing distillation range experimental results. The main interface of the Micro-Distillation Instrument software (MicroDistillAnalyzer_v1.0_setup) is shown in Figure 4.

#### 2.2.2. Raman Spectrometer Software

The main functions of the Raman spectrometer software include Raman spectroscopy data acquisition and analysis, drawing Raman spectra, and controlling the operation of the Raman spectrometer. The software used for the Raman spectrometer is BWSpec 4.10, which can transmit the read spectral data to a computer terminal for processing. The use of high-performance computers can improve the speed of data processing. After drawing the Raman spectrum, one can continue to use the computer for analysis or save the spectrum in the computer. The software interface of the Raman spectrometer is shown in Figure 5.

## 3. Data Processing Methods

In the actual operation of instruments, the original signal contains not only information about the chemical structure of the substance, but also noise generated by other interfering factors. Noise can affect the accuracy and repeatability of measurements, such as stray light noise in the optical path, dark current noise, and instrument assembly processes, which can cause a lot of noise interference in Raman spectra and produce many spikes on the spectral lines. In addition, Raman spectroscopy itself has a weak induction intensity, and noise can easily overwhelm the characteristic bands of the spectrum. In order to prevent the above-mentioned interference, this paper reduces these interference factors through data preprocessing methods, which improves the accuracy and stability of the calibration model and reduces errors.

### 3.1. Savitzy–Golay Smoothing Processing

This paper uses Savitzy–Golay smoothing method to smooth the spectrum. The smoothdata function that comes with the MATLAB 2108b software is used, with its parameters set to the ’sgolay’ method, which is smoothdata (raw_CPress, ’sgolay’). It performs smoothing based on the quadratic polynomial fitted on each window of A. When the data change rapidly, this method may be more effective than other methods and more suitable for the changing characteristics of Raman spectroscopy data. The default value for window width is 4.

Assuming a set of data in the spectral signal is *b*(*x*), *x* = −*M*, 0, *M*, smooth and denoise *b*(*x*) to obtain *y*(*x*).(1)$$y(x)=\sum_{k=-M}^{M}h\left(k\right)b(x-k)$$

*h*(*k*) is the sampling response of the S–G filter. After smoothing *y*(*x*), we can obtain:(2)$$y(x)=a_{0}+a_{1}x+a_{2}x^{2}+\cdots +a_{p}x^{p}p\leq 2M$$

The Raman spectrum obtained by connecting the curves of *y*(*x*) is the smoothed spectral curve, which removes the high-frequency segments in the Raman spectrum and fits the low-frequency segments in the Raman spectrum.

To obtain the *y*(*x*) polynomial, it is necessary to determine the coefficients of the polynomial. From Formula (2), we can know that *y*(0) = *a*_0_, so we only need to find the optimal combination *y*(0) of the window center point z(0), then translate the window to obtain its center point. We need to fit the residuals and use the least squares method to obtain:(3)$$\varepsilon =\sum_{l=-M}^{M}{\left(y(x)-b(x)\right)}^{2}=\overset{M}{\underset{l=-M}{\sum }}\left(\sum_{k=0}^{N}a_{k}x^{k}-b(x\right){)}^{2}$$

To minimize, the coefficients of Formula (3) are differentiated and made equal to 0, resulting in:(4)$$\frac{\partial \varepsilon }{\partial a_{i}}=\sum_{l=-M}^{M}{2x}^{i}(y(x)-b(x))=\sum_{l=-M}^{M}{2x}^{i}(\sum_{k=0}^{N}a_{k}x^{k}-Z(x))=0$$

By solving Formula (4), we obtain the coefficients of the polynomial and use the polynomial to represent the smoothed Raman spectrum.

### 3.2. Baseline Correction

The main reason for baseline shift in Raman spectroscopy is the fluorescence background present in the measured sample and sample container. The existence of a baseline can cause a shift in Raman spectra, affecting subsequent analysis of the spectrum, so it needs to be removed in advance. This paper chooses the least squares method as the baseline correction method, which is an improved version of the PLS method and adds roughness as a penalty term. In order to further reduce experimental errors, the number of experiments was increased. The induction intensity of characteristic bands was also increased, which resulted in a certain reduction in errors. After predicting and calculating distillation range and other indicators using PLS, we calculated the correlation coefficient of the corresponding spectral segment prediction results and optimized the selection of spectral segments based on the magnitude of the correlation coefficient. The baseline correction uses the Backcor function. This program estimates the background (or baseline) of optical spectra by the polynomial, minimizing a cost function [24,25].

### 3.3. Normalization Processing

In order to facilitate the processing of Raman spectra and enable the comparison of Raman spectra obtained at different integration times, the response intensity is normalized. Specifically, we first find the maximum and minimum values among all of the spectral intensities, subtract the minimum value from the spectral intensity of each point, and divide by the difference between the maximum and minimum values to obtain the relative intensity of the spectrum. We then replot all spectral relative intensity points to obtain the normalized spectrum. The calculation formula is as follows:(5)$$y_{i}^{*}=\frac{{y_{i}-y}_{min}}{{y_{max}-y}_{min}}$$

$y_{i}^{*}$ represents the spectral intensity at any point in the spectrum and $y_{max}$ and $y_{min}$ respectively represent the maximum and minimum values in the original Raman spectrum.

The normalized spectrum can be used for peak identification, calculating the intensity and position of the peaks. The specific method is to convert the Raman spectral intensity represented by the *y*-axis to logarithmic coordinates and fit the data of the newly generated coordinates (*x*, ln(*y*)) into a quadratic equation. The coefficients and constants of the quadratic term, first-order term, and constant term after quadratic fitting are *a*, *b*, and *c*. The formulas for calculating position and intensity can be obtained as follows:(6)P=−((D(2)×b/2c)−D(1))(7)$$H=exp(a-c{\left(b/2c\right)}^{2})$$

*P* represents the position of the peak, *H* represents the intensity of the peak, *D*(1) represents the average value on the *x*-axis, and *D*(2) represents the standard deviation on the *x*-axis.

## 4. Experimental Details

### 4.1. Materials

We collected 14 diesel samples from domestic and foreign oil depots, including 11 domestic diesel samples and 3 foreign diesel samples. The diesel sample numbers are shown in Table 1. Diesel oil (China National Petroleum Corporation Chongqing Marketing Branch, Chongqing, China). Anhydrous sodium sulfate (Aladdin Biochemical Technology Co., Ltd., Shanghai, China).

### 4.2. Experimental Methods

Add 10 g anhydrous sodium sulfate to a 100 mL sample, shake for 2 min, and let it stand for 15 min. After confirming the absence of water, take the upper clear liquid and store it between 1 °C and 10 °C for future use. Set the distillation time of the micro-distillation instrument to 8 min, the integration time of the Raman spectrometer to 30 s, and the power of the laser emitter to 100%, which is 340 mW. Set the average number of sample detections to 2 and spectral acquisition every 60 s. 10 mL of the sample is used for testing the joint detection system and 1.5 mL of the sample is directly used for Raman spectrometer testing. The working environment of both instruments is set at normal atmospheric pressure and room temperature.

### 4.3. Data Processing Experiment

Fill the 14 types of diesel oil samples into 10 mL and 1.5 mL quartz glass bottles. 10 mL of each sample is used for testing the joint detection system and 1.5 mL of each sample is directly used for Raman spectrometer testing. Repeat three times and take the average as the experimental result. The relative standard deviation (RSD) of the main peak intensity measured at the same point is 3–5%.

### 4.4. Joint Testing Experiment

The fluorescence interference of No. 1−10^#^ diesel oil 1 was strong and the fluorescence interference of No. 2−10^#^ diesel oil 2 itself was weak. The above two oil samples were used as the test objects. We put the undiluted sample into a 10 mL quartz glass bottle and directly placed it in a Raman spectrometer for testing. We repeated this procedure three times and took the average as the experimental result. During the sample distillation process, a joint detection system was used for testing. Eight samples were collected in the order of their distillation and numbered as 1–8. We collected Raman spectra using the same integration time setting. We repeated this procedure three times and took the average as the experimental result.

## 5. Results and Discussion

### 5.1. Data Smoothing

The data processing results of this joint detection system are shown in Figure 6. Figure 6a shows the original Raman spectra of 14 diesel samples, from which a small number of burrs can be observed. With reference to a literature comparison [27,28], the results of this experiment conform to the characteristics of Raman spectroscopy. The result of smoothing the spectrum using the Savitzy–Golay smoothing method is shown in Figure 6b. The characteristic bands of the processed spectral lines are clearer, smoother, have fewer burrs, and are easier to analyze. The results confirmed that Savitzy–Golay smoothing data processing is suitable for this contact detection system.

Analyzing the data processing mechanism of this measurement system, the procedure is to select a window with a certain width in the Raman data signal and introduce polynomial least squares fitting in the window moving operation. This method achieves good noise reduction, reducing noise interference in the spectrum while preserving useful information in the analysis signal. Its effect is to remove burr interference while maximizing the retention of Raman spectral characteristic bands, which is of great significance for further data processing research of this joint detection system.

### 5.2. Baseline Correction and Normalization Processing

The Raman spectrogram of this joint detection system after baseline correction is shown in Figure 7a, where the positions of characteristic bands are more obvious, but the intensity of smaller characteristic bands is weaker. Therefore, the advantage of using the least squares method is that it does not require prior knowledge of the peak shape and baseline for baseline correction calculation, saving preprocessing steps. It is also fast in fitting calculations, which is beneficial for adjusting it. But when the peak area is small, there is significant interference, which affects the fitting results.

The normalized Raman spectrogram of this joint detection system is shown in Figure 7b. It can be seen that the positions of the characteristic bands are clearer and the intensities are more obvious, which means that the system can better perform spectrum analysis and obtain the composition and content of the sample. This is mainly because the normalization process replots the relative intensity points of all of the spectra, resulting in a normalized spectrum. Especially for bands with smaller characteristic bands, our system can better identify spectral bands and calculate their intensities and positions.

### 5.3. Joint Testing System Test Results

Taking No. 1−10^#^ diesel oil 1 as the test object, the Raman spectrum of the undiluted sample is shown in Figure 8a. The fluorescence interference of this sample is severe, and the characteristic bands are submerged. The detection results of the distillate during the distillation process are shown in Figure 8b. According to the order of the distillate, the Raman spectra curves of the distillate in the early stage of distillation are numbered 1–7. Clear and distinguishable characteristic bands can be seen. The fluorescence interference of the final distillate sample No. 8 is still strong, without obvious characteristic bands, and its characteristic peak interference is similar to the Raman spectrum of the undiluted sample. The above results indicate that there are substances that exist in small amounts in the light components of petroleum products, which can cause Raman spectral fluorescence interference. The substances that have a significant impact on the characteristic bands of Raman spectra mainly exist in the heavy components of petroleum products.

Taking No. 2−10^#^ diesel oil 2 as the test object, the Raman spectrum of the undiluted sample is shown in Figure 9a. The sample has relatively less fluorescence interference and there is no significant difference in characteristic bands compared to the detection results of the distillate during the distillation process, as shown in Figure 9b. The above results indicate that for diesel samples with weak or no fluorescence interference, the spectral characteristic bands are almost unaffected before and after distillation. However, it has a certain correction effect on raising the original spectral baseline and a certain reduction in sensitivity to changes in integration time.

## 6. Conclusions

This paper mainly designs a small-scale oil joint detection system. The joint detection system combines a micro-distillation instrument and a Raman spectrometer module through a combination module. Based on the actual usage of Raman spectroscopy, the Savitzy–Golay smoothing method is selected to penalize the baseline correction and normalization of the least squares method. The least squares method is used to establish a quantitative analysis multivariate correction model, linking the spectral data of the tested substance with its composition and properties. This paper selects 14 different sources and grades of diesel for pre-experiments to verify the data fitting effect. Two oil samples, No. 1−10^#^ diesel oil 1 with strong fluorescence interference and No. 2−10^#^ diesel oil 2 with weak fluorescence interference, were used as test objects to compare the Raman spectra of undiluted oil and distilled oil, verify the testing effect of the joint detection system, and analyze the mechanism of fluorescence interference mainly existing in the heavy components of the oil. The above results verify the effectiveness of the combined detection system for measuring trace distillation and Raman spectroscopy. This system has laid the foundation for research related to distillation range combined with Raman spectroscopy technology.

## Data Availability

Data are contained within the article.

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
