# Peer review of "Research on a Small Oil Detection Instrument Combining Micro-Distillation Range Analyzer and Raman Spectroscopy Technology"

_sensors, 2025, doi:10.3390/s25072130_

Round 1
Reviewer 1 Report
Comments and Suggestions for Authors
The manuscript is quite interesting, especially for the idea of using a miniaturized system made by the authors. The work is well-written, however, some important information is missing.
Although overall it is well-written, the results section is lacking. There is no characterization of the Raman bands for the different types of diesel oil, at least some comparison with literature data, so that readers can confirm that it is indeed a diesel oil spectrum.
The article promises an evaluation of the oil composition and even a "quantification", but there are no results on this part. Only the spectrum treatments are shown, and the normalization was not very good! The authors need to define whether they will only show the use of the miniaturized system and the treatments involved in it (for this, the optimization process needs to be added to find the best parameters for each treatment, maybe a SI file?), there is also no mention of which parameters were best for each treatment. Or if the application promised in the methodology will also be included.
In the attached PDF, the more specific questions are marked for better visualization by the authors.

Reviewer 2 Report
Comments and Suggestions for Authors
This paper describes an apparatus for simultaneous distillation and Raman analysis of diesel oils. The device is innovative and is surely interesting, because it would allow monitoring in real time distillation of diesel oil and evaluation of its quality.
Description of the device is satisfactory and is accompanied by both photos and schemes. Description of data processing procedure is acceptable, despite some unclear points that will be described below in detail.
The weak point of the paper is the presentation of results, which would be worth an expansion. In the paper the authors describe a procedure for estimation of position and heights of Raman peaks. In chapter 5, however, no such calculations are presented for the oils of their dataset, nor any correspondence between Raman peaks and components of diesel oil is discussed.
Presentation of results is mainly focused on the outcome of pre-processing methods and the change of fluorescence intensity during distillation of just 2 out of 14 oils of the dataset. There is no discussion about parameters determining the quality of diesel oils and their relationship with the Raman spectrum. Moreover, without knowing in what differ the 14 analyzed samples it impossible to judge if the experimental design is appropriate.
Addressing this point would surely add value to the paper, making it more interesting. For this reason, I would suggest a major revision.
Aside from this major point, some minor corrections would be advisable, which are listed below.
- The caption of Figure 5 incorrectly identifies it ad “Figure 4”, please correct.
- Paragraph 3.1: Parameters of the Savitsky-Golay filter used should be given, such as the degree of the fitting polynomial and the width of the smoothing window.
- Paragraph 3.2: The baseline correction method should be described in detail, and a reference about it should be added to the paper. Authors say that PLS was used but, as far I know,the meaning of PLS is “Partial Least Square”, which is a multivariate regression tool. I do not understand how it can be used to fit a spectrum baseline. If this is the case, such procedure should be explained in the text. If, alternatively, the acronym PLS has a different meaning (es. Polynomial Least Square) it should be explicitly stated in the text.
- Paragraph 4.1: Please explain the meaning of 10# and 0# notation in samples names. It seems to be a technical petrochemical code which could not be known to all readers. It should also be advisable does not split Table 1 between two different pages.
- Figures 8 and 9: Please increase the size legend, because it is not readable.
Round 2
Reviewer 1 Report
Comments and Suggestions for Authors
File attached.

Reviewer 2 Report
Comments and Suggestions for Authors
The answers of authors to my questions and the amendments to the paper are satisfactory. Therefore, for me, the publishing procedure can go on.
Author Response
Thank you very much.